# Acetyl-L-Carnitine and Liposomal Co-Enzyme Q_10_ Attenuate Hepatic Inflammation, Apoptosis, and Fibrosis Induced by Propionic Acid

**DOI:** 10.3390/ijms241411519

**Published:** 2023-07-15

**Authors:** Ahlam M. Alhusaini, Rahaf Alsoghayer, Lina Alhushan, Abeer M. Alanazi, Iman H. Hasan

**Affiliations:** 1Department of Pharmacology and Toxicology, College of Pharmacy, King Saud University, P.O. Box 22452, Riyadh 11459, Saudi Arabia; aalanazi22@ksu.edu.sa; 2Pharm D Program, College of Pharmacy, King Saud University, P.O. Box 22452, Riyadh 11459, Saudi Arabia; 437201593@student.ksu.edu.sa (R.A.); 438201662@student.ksu.edu.sa (L.A.)

**Keywords:** propionic acid, acetyl-L-carnitine, liposomal-coenzyme Q_10_, cytokeratin-18, transforming growth factor-β1

## Abstract

Propionic acid (PRA) is a metabolic end-product of enteric bacteria in the gut, and it is commonly used as a food preservative. Despite the necessity of PRA for immunity in the body, excessive exposure to this product may result in disruptive effects. The purpose of this study is to examine the hepatoprotective effects of acetyl-L-carnitine (A-CAR) and liposomal-coenzyme Q_10_ (L-CoQ_10_) against PRA-induced injury. Liver injury in rats was induced by oral administration of PRA, and A-CAR and L-CoQ_10_ were administered concurrently with PRA for 5 days. Oxidative stress, inflammatory, apoptotic, and fibrotic biomarkers were analyzed; the histology of liver tissue was assessed as well to further explore any pathological alterations. PRA caused significant increases in the levels of serum liver enzymes and hepatic oxidative stress, inflammatory, and apoptotic biomarker levels, along with histopathological alterations. Concurrent treatment with A-CAR and/or L-CoQ_10_ with PRA prevented tissue injury and decreased the levels of oxidative stress, proinflammatory cytokines, and apoptotic markers. Additionally, A-CAR and/or L-CoQ_10_ modulated the expression of high-mobility group box-1, cytokeratin-18, transforming growth factor-beta1, and SMAD3 in liver tissue. In conclusion, A-CAR and/or L-CoQ_10_ showed hepatoprotective efficacy by reducing oxidative stress, the inflammatory response, apoptosis, and fibrosis in liver tissue.

## 1. Introduction

Propionic acid (PRA) is a naturally occurring carboxylic acid and it is commonly used as a food preservative due to its antimicrobial activity [1]. The probiotic effect of PRA has drawn some interest, as an appropriate amount of PRA improves the immune system and enhances cell signaling [2]. However, the continuous intake of PRA may cause medical concerns as its long-term effects are still unknown. A study in Singapore examined the effect of overexposure to PRA; they compared people receiving a placebo with people consuming daily bread with 2% added PRA, and they found that PRA caused irritability, restlessness, inattention, and sleep disturbances [3]. As the axis transporting PRA to the brain passes through the liver, a high dose of PRA could cause severe liver injury [4]. The liver is an important organ in the body; it acts as a regulatory region for multiple physiological processes, including the metabolism of carbohydrates and fat, the synthesis of bile acids, and the chemical detoxification of several drugs [5]. The antioxidant activity in hepatic tissue occurs in a programmed manner to prevent the progression of oxidative stress; however, if the antioxidant capacity is overwhelmed, molecular damage can be produced. Thus, there is a need for an ideal antioxidant agent to prevent the excessive oxidative stress induced by PRA overdose.

L-carnitine is a chemical product in the human body that is derived from the essential amino acid lysine through a specific biosynthesis pathway; it is synthesized naturally in the brain, liver, and kidneys [6]. L-carnitine deficiency is implicated in several diseases, including cardiomyopathy, diabetes, and cirrhosis [7]. Additionally, it can be found in some foods, such as poultry, meat, fish, and dairy products, and it is available as supplements in ester form as acetyl-L-carnitine (A-CAR) [7].

Coenzyme Q_10_ is synthesized naturally in the body, and it exists in most aerobic organisms [8]. The most important function of CoQ_10_ is its involvement in the electron transport chain (ETC), which is responsible for adenosine triphosphate (ATP) production; hence, CoQ_10_ is an energy transfer molecule. Accordingly, it is found most notably in body organs that require high energy production, such as the heart, kidneys, muscles, and liver. Furthermore, CoQ_10_ functions as a fat-soluble antioxidant that prevents reactive oxygen species (ROS) from causing damage to cell proteins, lipids, and DNA [8]. Due to the poor bioavailability of CoQ_10_ supplements, a liposomal formulation of CoQ_10_ (L-CoQ_10_) has been developed to enhance the cellular uptake of the active drug without degradation [9].

This study aims to examine the protective effects of A-CAR and/or L-CoQ_10_ against PRA-induced liver injury by assessing the oxidative stress, inflammatory, apoptotic, and fibrotic mechanisms.

## 2. Results

### 2.1. A-CAR and/or L-CoQ_10_ Retained Liver Function in PRA-Induced Liver Injury

The group of rats exposed to PRA had significantly higher serum AST, ALT, and LDH enzyme activity levels than the control group. The concurrent administration of A-CAR and/or L-CoQ_10_ with PRA showed a marked reduction in these enzymes when compared with the PRA group (Figure 1). Moreover, the groups treated with A-CAR monotherapy showed a significant reduction in the level of LDH when compared with the L-CoQ_10_ monotherapy or the combined therapy with both agents (Figure 1).

### 2.2. A-CAR and/or L-CoQ_10_ Attenuated Oxidative Stress Induced by PRA

Measuring the levels of oxidative markers provides an assessment of liver condition. The administration of PRA resulted in a significant elevation in MDA levels and a reduction in the endogenous antioxidant capacity levels of GSH and SOD, indicating high oxidative damage (Figure 2). The administration of A-CAR and/or L-CoQ_10_ significantly attenuated the oxidative stress in liver tissue by reducing MDA levels and restoring the level of GSH and the activity of the SOD enzyme (Figure 2). Interestingly, the combined treatment significantly reduced the MDA level compared to the A-CAR group (Figure 2).

### 2.3. A-CAR and/or L-CoQ_10_ Diminished the Inflammation and Apoptosis Induced by PRA

As a result of the administration of PRA, the expression of the proinflammatory biomarkers IL-6 and TNF-α was increased compared with the control group, whereas they were downregulated in the groups treated with A-CAR and/or L-CoQ_10_ (Figure 3). Moreover, the administration of PRA triggered apoptosis by causing a remarkable increase in caspase-3 expression in the liver. In contrast, the groups treated with A-CAR and/or L-CoQ_10_ exhibited lower caspase-3 levels compared with the PRA group (Figure 3). The combined therapy with both agents markedly diminished the levels of inflammatory and apoptotic markers compared to the L-CoQ_10_ group (Figure 3).

### 2.4. A-CAR and/or L-CoQ_10_ Suppressed the Hepatic Expression of TGF-β and SMAD3 Proteins

The administration of a high dose of PRA caused the upregulation of fibrotic markers (TGF-β1 and SMAD3) at the protein level in hepatic tissue compared to the normal control group; however, treatment with A-CAR and/or L-CoQ_10_ significantly downregulated the expression levels of these proteins (Figure 4). The group treated with both agents showed a significant reduction in the expression levels of TGF-β1 (Figure 4).

### 2.5. A-CAR and/or L-CoQ_10_ Downregulated the Hepatic Expression of HMGB1 and CK18 mRNA

PRA considerably upregulated the expression levels of HMGB1 and CK18 mRNA in liver tissue when compared to the healthy group (Figure 5). The groups that received A-CAR and/or L-CoQ_10_ showed significant downregulation in the gene expression of HMGB1 and CK18 compared to the PRA group (Figure 5). The combination of both drugs showed a remarkable effect when compared with the monotherapies (Figure 5).

### 2.6. A-CAR and/or L-CoQ_10_ Improved the Morphological Changes Induced by PRA

The histopathological examination of liver sections from the normal control group revealed normal histological features (Figure 6A), while liver sections from the PRA-intoxicated group displayed hepatocytes with intracytoplasmic vacuoles, congested and dilated sinusoids, and necrotic areas (Figure 6B). Treatment with A-CAR improved the liver injury induced by PRA; the hepatocytes were arranged in thin plates with binucleated nuclei (Figure 6C). Moreover, the liver sections from the L-CoQ_10_-treated group showed an intact lobular hepatic architecture, with hepatocytes arranged in thin plates with mild hydropic degeneration (Figure 6D). As expected, the combined therapy with L-CoQ_10_ and A-CAR produced an additive effect, allowing the prevention of hepatic cell injury, with an almost normal structure and architecture (Figure 6E).

## 3. Discussion

PRA is a short-chain fatty acid found in the gut; it acts as the end metabolic product of enteric bacteria [10]. It is essential for the functioning of immunological and physiological processes [2]. Nonetheless, abnormal levels of PRA lead to undesirable effects, such as mitochondrial dysfunction and oxidative stress, as well as metabolic or immune reactions [11]. This work was designed to investigate the possible protective effects of A-CAR and L-CoQ_10_ against PRA-induced hepatic damage by assessing specific oxidative stress, inflammatory, apoptotic, and fibrotic biomarkers. A-CAR is an acetylated derivative of carnitine; it has anti-inflammatory and antioxidant properties [7]. CoQ_10_ is a lipid-soluble compound that works as a natural scavenger of free radicals; additionally, it acts on nuclear factor kappa B (NF-kB)-dependent gene expression and it suppresses the gene expression of TNF-α [12]. A nanoparticle that targets specific organs represents an important strategy in drug delivery systems; liposomes with a nanoparticle size are composed of phospholipids that have the ability to deliver their contents into the cell cytoplasm with fewer side effects [13].

In the present study, liver injury induced by PRA was manifested by the significant elevation of the serum levels of liver function biomarkers, including ALT, AST, and LDH activity. Similar effects have been demonstrated in a previous study using PRA-induced liver injury in a rat model [4]. A-CAR and L-CoQ10, as well as their combination, diminished the hepatotoxic effects of PRA, as evidenced by the significant reduction in ALT, AST, and LDH activity levels and decreased hepatic degeneration in the histopathological results, which was in line with earlier studies [12,14,15,16]. Mohamed et al. (2019) found that the treatment of rats with CoQ_10_ protected against liver injury, which was manifested by decreased levels of ALT and AST [17]. In addition, a previous meta-analysis by Hyunwoo and colleagues (2022) reported a beneficial effect of L-carnitine in reducing the aforementioned liver enzymes in patients with chronic liver disease, similar to our findings [18].

From our biochemical findings, PRA induced oxidative stress, as proven by the decreasing levels of antioxidant biomarkers GSH and SOD and increased MDA levels. Oxidative stress is one of the multiple mechanisms involved in PRA-induced liver toxicity [19]. Moreover, a recent study by Erten has reported that PRA increases ROS generation mediated by polyunsaturated fatty acid oxidation and produces MDA, which is responsible for biomolecular organ damage [20]. In the current study, both A-CAR and L-CoQ_10_ significantly attenuated PRA-induced oxidative stress by improving the activity of the SOD enzyme and the levels of endogenous GSH and reducing MDA levels. These results are in parallel with the findings of previous studies that proved the antioxidant effects of A-CAR and L-CoQ_10_ [21,22]. L-carnitine works as a protective agent against lipid peroxidation by decreasing the formation of hydrogen peroxide, and it increases the antioxidant effect by preventing the depletion of GSH, SOD, and CAT, as well as improving the free radical scavenging activity [23]. Furthermore, the protective roles of A-CAR as an antioxidant can be explained by an indirect mechanism, as it inhibits the activation of free radical production pathways such as iNOS and NOX [24].

Besides oxidative stress, inflammation has been implicated in the hepatotoxicity induced by PRA, as it has been shown that it can upregulate TNF-α and increase the levels of inflammatory cytokines [25]. In accordance with the previously mentioned data, in our results, we revealed that PRA caused the significant elevation of TNF-α and IL-6 levels, while these inflammatory cytokines’ levels were significantly decreased after the A-CAR and L-CoQ_10_ interventions. In a study published in 2020 by Wang et al., they reported that A-CAR reduced the levels of inflammatory factors TNF-α and IL-1β in a rat model of atherosclerosis [21]. Similarly, Bodaghi-Namileh and his assistants documented that IL-6 expression was significantly decreased following the administration of A-CAR in arsenic-induced liver injury [26], and these inflammatory markers declined in patients with acute ischemic stroke after A-CAR treatment [27]. Moreover, the proposed mechanisms behind the anti-inflammatory effects of A-CAR were recently investigated; it significantly alleviated the levels of mRNA and the protein expression of C-reactive protein (CPR), TNF-α, and IL-1β and attenuated inflammatory angiogenesis by reducing triggered endothelial cells and macrophage infiltration, as well as chemokine receptor type 4 (CXCR-4) [28,29].

Additionally, CoQ_10_ affects TNF-α and IL-1β gene expression in diabetic nephropathy patients, mostly through the NF-κB transcriptional pathway; likewise, in a previous systematic review in 2019 on patients with chronic inflammatory diseases, it was revealed that CoQ_10_ supplementation diminished the serum levels of TNF-α and IL-6 [22,30]. While the anti-inflammatory effect of CoQ_10_ depends on its ability to regulate the gene expression of IL-1 and IL-6, as well as TNF-α, CoQ_10_ reduces the production of proinflammatory cytokines by affecting the gene expression of nuclear factor kappa B (NF-κB). Other researchers suggest that the anti-inflammatory effect of CoQ_10_ may be attributed to adiponectin (which is a protein with an anti-inflammatory effect); CoQ_10_ supplementation leads to an increase in the levels of the adiponectin hormone, which consequently causes the depletion of the inflammatory action mediated by TNF-α [22]. Similarly, CoQ_10_ administration in patients with migraine significantly reduced the levels of calcitonin gene-related peptide (CGRP), which is an important inflammatory mediator [31].

In addition to the inflammatory action of PRA, this study examined the effect of PRA on the expression levels of HMGB1 mRNA. HMGB1 is one of the danger-associated molecular patterns (DAMPs) that has dual activity inside and outside the cells. Intracellularly, it acts in the nucleus and contributes to the transcription of the gene [32]. Extracellularly, it contributes to the inflammation process. It is released in the extracellular space following endogenous danger triggers such as infection [32,33]. A recent study demonstrated high serum levels of HMGB1 in patients with multiple sclerosis [34]. Moreover, in a mouse model of malaria-associated acute lung injury, both plasma and lung tissue showed high levels of HMGB1 [32]. Here, the results of the gene analysis indicated that PRA increased the expression of HMGB1 mRNA, whereas A-CAR and L-CoQ_10_ as well as their combination significantly attenuated the HMGB1 expression levels. The damaged liver releases HMGB1, which binds to its receptors, and this subsequently activates signaling pathways, leading to the secretion of proinflammatory cytokines including TNF-α, which causes acute liver injury [34,35]. To our knowledge, this is the first study examining the effect of PRA, A-CAR, and L-CoQ_10_ on the expression levels of HMGB1 mRNA.

Additionally, oxidative stress plays an essential role in apoptosis; it disrupts the mitochondrial membrane potential and consequently leads to the release of pro-apoptotic proteins [36,37]. CK18 represents approximately 5% of the total proteins in the liver and plays an important role in apoptosis; thus, it can serve as a marker of cell death [38]. During apoptotic processes, CK18 is released from the hepatic cells, and it is cleaved by effector caspases that facilitate the formation of apoptotic bodies and augment the apoptotic signal [39]. In this work, PRA caused the upregulation of caspase-3 and CK18 expression levels. In contrast, the concomitant administration of A-CAR and L-CoQ_10_ significantly attenuated the expression levels of these proteins. The findings of the current study support our previous study that revealed the antioxidant and antiapoptotic properties of A-CAR and L-CoQ_10_ in the brain tissue in a rat model of autism [40]. To the best of our knowledge, this is the first study investigating the effect of PRA, A-CAR, and L-CoQ_10_ on CK18 expression levels. A-CAR has antiapoptotic activity that is mediated by the induction of the X-chromosome-linked inhibitor of apoptosis protein (XIAP) and affects all steps of the apoptotic cascade, such as inhibiting cytochrome c cytosolic release and DNA fragmentation. Likewise, in a H_2_O_2_-induced oxidative stress model, CoQ_10_ treatment significantly decreased the expression of the proapoptotic proteins Bax and caspase-3 and increased the expression of the antiapoptotic protein Bcl-2. Moreover, a previous study suggested that CoQ_10_ exerts an antiapoptotic effect by affecting the Nrf-2/NQO-1 signaling pathway [41,42].

To further explore the protective mechanisms of A-CAR and/or L-CoQ_10_ on PRA hepatoxicity, the expression of TGF-β1 and SMAD3 proteins was determined. Sustained liver inflammation produces structural changes that lead to fibrosis [36,43]. The TGF-β1 cytokine plays an important role in different biological processes, such as neoplastic growth and cell proliferation as well as apoptosis. The upregulation of TGF-β1 expression is a consistent feature of most fibrotic diseases, Moreover, SMAD is a central intracellular effector protein of TGF-β1 [34,35,44,45]. In the current study, TGF-β1 and SMAD3 protein levels were observed to be significantly increased in rats treated with PRA; on the other hand, the levels of these proteins were restored to normal levels following the administration of A-CAR and L-CoQ10, compared to the PRA-treated rats. Recent data showed that CoQ_10_ administration ameliorated the inflammation and fibrosis induced by radiation by downregulating the NF-kB/TGF-β1 pathway [36,43].

From our experiment, we concluded that A-CAR and/or L-CoQ_10_ showed a protective effect against PRA-induced liver damage. The possible mechanisms of CAR and/or L-CoQ_10_ hepatoprotection might be a reduction in oxidative stress, inflammation, and apoptosis, as well as downregulating the protein expression of fibrotic markers such as TGF-β1 and SMAD3. Furthermore, HMGB1 and CK18 could be promising drug targets for many disorders, especially those triggered by inflammation and apoptosis.

## 4. Materials and Methods

### 4.1. Chemicals

PRA and A-CAR were purchased from Sigma-Aldrich Co. L-CoQ_10_ with a diameter of less than 200 nm was obtained from LipoLife, Drakes Lane Industrial Estate, Drakes Lane, (UK). Alanine transaminase (ALT), aspartate transaminase (AST), and lactate dehydrogenase (LDH) kits were purchased from EKF Diagnostics, Inc., South Bend, IN, USA. Tumor necrosis factor-α (TNF-α), interleukin-6 (IL-6), and caspase-3 ELISA kits were purchased from My BioSource^®^ (San Diego, CA, USA).

### 4.2. Animals

Thirty adult male albino rats weighing 150–170 g were obtained from the Experimental Animal Center, King Saud University. The animals were allowed to adapt to the laboratory for one week under standard environmental conditions with a temperature of 22 °C and a natural light/dark cycle. They were given a standard rat pellet diet and distilled water ad libitum. The use of animals and the experimental design were approved by the Scientific Research Ethics Committee, King Saud University (KSU-SE-19-15).

### 4.3. Experimental Design

Rats were divided into five groups, with six rats each. Group I (control): rats received 1% carboxymethylcellulose; Group II: rats were orally intoxicated with PRA at a dose of 250 mg/kg/day [46]; Group III: rats were treated with A-CAR 100 mg/kg/day [47] at one hour after PRA administration; Group IV: rats were orally administered 10 mg/kg/day of L-CoQ_10_ [48] one hour after PRA intoxication; Group V: PRA-intoxicated rats were orally administered a mixture of A-CAR and L-CoQ_10_ at the same doses one hour after PRA administration. All treatments were given at a volume of 0.5 mL along with PRA for five days.

After the completion of the experiment, all rats were euthanized using CO_2_ and killed by decapitation. Blood samples were collected for serum separation by centrifugation at 3000 rpm at 4 °C for 15 min. Liver tissue was collected and then homogenized in phosphate-buffered saline solution to yield 20% homogenates. The homogenates were centrifuged at 3000 rpm at 4 °C for 30 min, and the supernatants were collected for further analysis. Liver samples from different animals were kept under nitrogen for Western blot analysis. The other liver samples were kept for histological examination in 10% formalin.

### 4.4. Determination of the Serum Levels of Liver Function Enzymes

ALT, AST, and LDH were measured using commercially available colorimetric kits, according to the manufacturer’s instructions.

### 4.5. Assessment of Oxidative Stress Biomarkers in Liver Tissue

Malondialdehyde (MDA) was assayed in the hepatic homogenates via the thiobarbituric acid reaction, according to the method of Ohkawa and colleagues [49]. A reduced glutathione (GSH) level was determined using the colorimetric technique of Ellman [50], and the superoxide dismutase (SOD) enzyme activity in liver tissue was assessed using the method of Marklund [51].

### 4.6. Measurement of Inflammatory and Apoptotic Biomarkers in Liver Tissue

The levels of TNF-α, IL-6, and caspase-3 in hepatic homogenates were measured quantitively according to the manufacturer’s specifications, using the ELISA sandwich immunoassay kit.

### 4.7. Determination of the Expression of TGF-β1 and SMAD3 Proteins

Western blots were performed to determine the protein expression of transforming growth factor- β1 (TGF-β1) and SMAD3. The samples were homogenized in RIPA buffer and proteinase inhibitors, and then centrifuged; the supernatant was collected, and its protein content was estimated using the Bradford protein assay kit (BioBasic, Markham, ON, Canada). Sixty μg protein was subjected to 10% SDS/PAGE followed by electro-transfer to nitrocellulose membranes. The membranes were incubated with mouse anti-TGF-β1 (Santa Cruz, CA, USA, Cat. No. sc-52893), mouse anti-SMAD3 (Santa Cruz, CA, USA; Cat. No. sc-101154), and mouse anti-β-actin (Santa Cruz, CA, USA; Cat. No. sc-8432) overnight at 4 °C. Following washing, the membranes were probed with rabbit anti-mouse secondary antibody (Santa Cruz, CA, USA, Cat. No. SC-358914) and then developed using the BCIP/NBT substrate detection kit (GeneMed Biotechnologies, Torrance, CA, USA). Protein bands were visualized using the ECL-Plus identification system (Amersham Life Sciences, Little Chalfont, Buckinghamshire, UK), according to the manufacturer’s instructions. Positive immunoreactive bands were quantified densitometrically and compared with the control.

### 4.8. Determination of the Expression of HMGB-1 and CK18 mRNA

The mRNA abundance of high-mobility group box-1 (HMG-1) and cytokeratin-18 (CK18) in the liver tissue of rats was determined using qRT-PCR. The isolated RNA was measured and samples with A260/A280 ≥1.8 were nominated for reverse transcription into cDNA. Synthesis of cDNA was accomplished using a high-capacity cDNA reverse transcription kit, and the obtained cDNA was amplified using the Maxima SYBR Green/ROX qPCR master mix and the primers in Table 1.

### 4.9. Histopathological Analysis

Liver samples were fixed in 10% formaldehyde, and thinly sliced sections were used for histopathological examination using hematoxylin and eosin (H&E) staining.

### 4.10. Statistical Analysis

Data were expressed as mean ± SEM for quantitative measures. The statistical comparisons were performed using one-way analysis of variance (ANOVA), followed by the Tukey–Kramer multiple-comparisons post-hoc test. The levels of significance were set at *p* < 0.05, *p* < 0.01, and *p* < 0.001. Statistical tests were conducted using GraphPad Prism 5.00 (GraphPad Prism, San Diego, CA, USA).

## Figures and Tables

**Figure 1 ijms-24-11519-f001:**
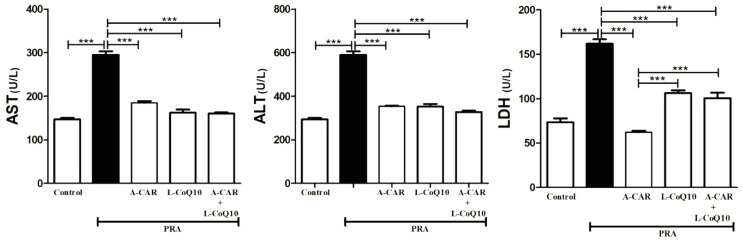
Treatment with A-CAR and L-CoQ_10_ either alone or in combination ameliorated serum AST, ALT, and LDH levels. Data are expressed as mean ± SEM (*n* = 6). *** *p* < 0.001.

**Figure 2 ijms-24-11519-f002:**
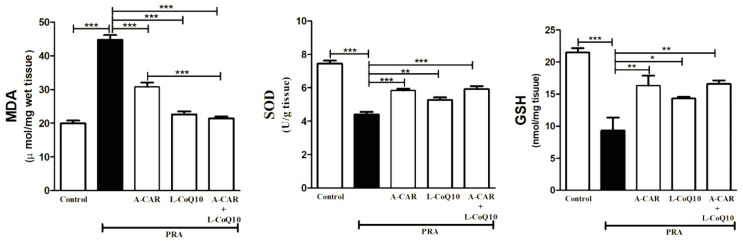
Treatment with A-CAR and L-CoQ_10_ either alone or in combination decreased hepatic MDA levels and increased SOD activity and GSH levels. Data are expressed as mean ± SEM (*n* = 6). * *p* < 0.05, ** *p* < 0.01, *** *p* < 0.001.

**Figure 3 ijms-24-11519-f003:**
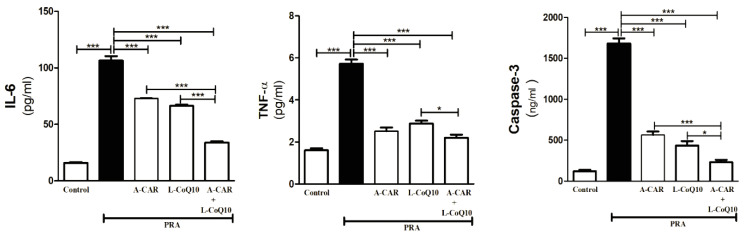
Treatment with A-CAR and L-CoQ_10_ either alone or in combination decreased hepatic inflammatory biomarkers (IL-6 and TNF-α) and apoptotic biomarker (caspase-3). Data are expressed as mean ± SEM (*n* = 6). * *p* < 0.05, *** *p* < 0.001.

**Figure 4 ijms-24-11519-f004:**
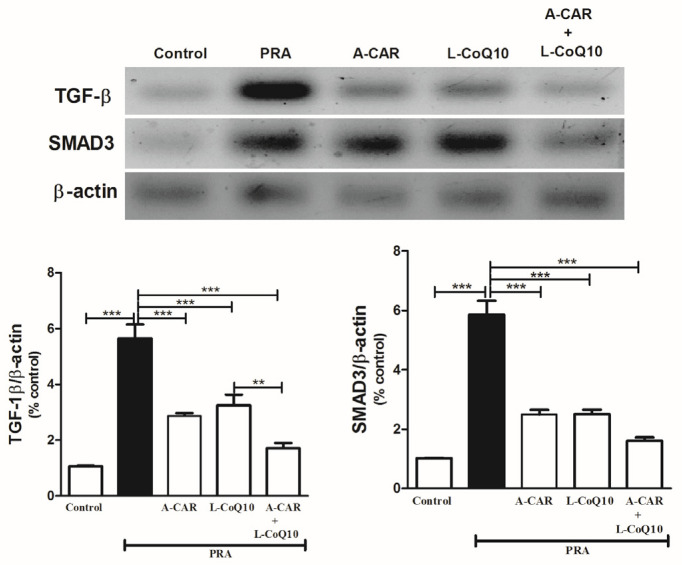
A-CAR and L-CoQ_10_ suppressed hepatic expression levels of TGF-β and SMAD3 proteins in PRA-intoxicated rats. Representative blots show the changes in the expression of TGF-β and SMAD3 proteins in control and all treated groups. Data are expressed as mean ± SEM (*n* = 6). ** *p* < 0.01, *** *p* < 0.001.

**Figure 5 ijms-24-11519-f005:**
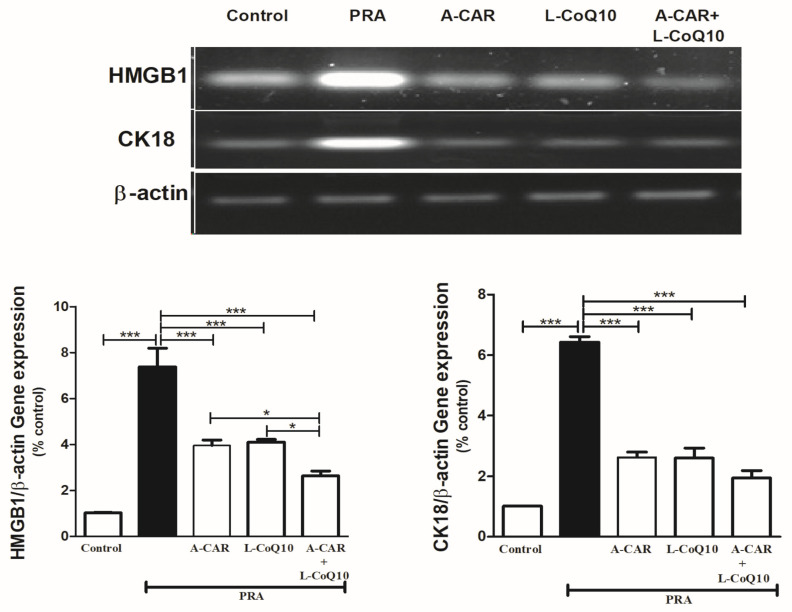
A-CAR and L-CoQ_10_ downregulated the hepatic expression of HMGB1 and CK18 mRNA in PRA-intoxicated rats. Representative RT-PCR gel electrophoresis shows variations in the expression levels of HMGB1 and CK18 mRNA in control and all treated groups. Data are expressed as mean ± SEM (*n* = 6). * *p* < 0.05, *** *p* < 0.001.

**Figure 6 ijms-24-11519-f006:**
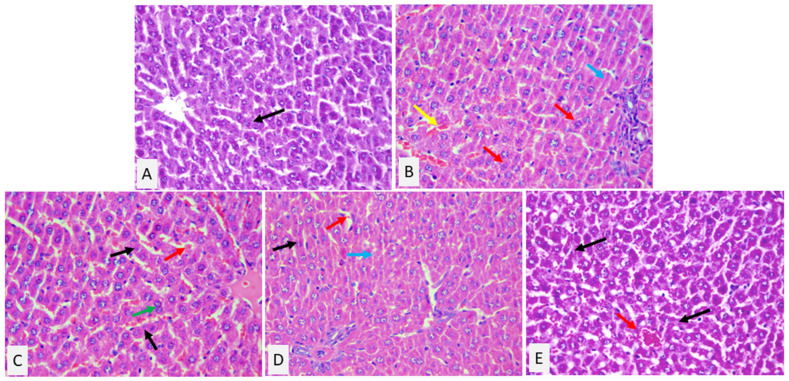
Histology of liver tissue stained with H&E, x400. (**A**) Liver tissue of control group with normal histological architecture and hepatocytes arranged in thin plates (black arrow). (**B**) Liver tissue of PRA-intoxicated group shows hepatocytes with intracytoplasmic vacuoles (red arrow), congested and dilated sinusoids (yellow arrow), and necrotic areas (blue arrow). (**C**) Liver tissue of A-CAR-treated group shows hepatic tissue with relatively normal structure, hepatocytes arranged in thin plates (black arrow), congested sinusoids (red arrow), and hepatocytes with binucleated nuclei (green arrow). (**D**) Liver section of L-CoQ_10_-treated group shows intact lobular hepatic architecture, hepatocytes arranged in thin plates with mild hydropic degeneration (black arrow), congested sinusoids (red arrow), and focal area of necrosis (blue arrow). (**E**) A representative liver section of the group treated with the combination of L-CoQ_10_ and A-CAR shows hepatic tissue with an almost normal structure and architecture, hepatocytes arranged in thin plates (black arrow), and a congested central vein (red arrow).

**Table 1 ijms-24-11519-t001:** Primer sequences used in the quantitative RT-PCR.

*HMGB-1*	**Forward**
	5′-TTGTCCACACACCCTGCATA-3′
	**Reverse**
	3′-AATTGATCACTCCTTGCTTTGCT-5′
*CK18*	**Forward**
	5′-GAGACGTACAGTCCAGTCCTTGG-3′
	**Reverse**
	3′-CCACCTCCCTCAGGCTGTT-5′

## Data Availability

The data presented in this study are available on request from the corresponding author.

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
