# Peer review of "Acetyl-L-Carnitine and Liposomal Co-Enzyme Q10 Attenuate Hepatic Inflammation, Apoptosis, and Fibrosis Induced by Propionic Acid"

_ijms, 2023, doi:10.3390/ijms241411519_

Round 1
Reviewer 1 Report
This study demonstrated that “Acetyl-L-carnitine and liposomal co-enzyme Q10 attenuate hepatic inflammation, apoptosis, and fibrosis induced by propionic acid”. This manuscript is very interesting, however, there are several concerns relating that should be carefully address by the authors.
Materials and Methods
How did authors decide the concentrations of PRA (250 mg/kg/day), A-CAR (100 mg/kg/day) and L-CoQ10 (10 mg/kg/day) for dose to rats?
Authors described as “All treatments were given along with PRA for five days.”
All samples were orally administered; however, authors did not describe timing of administration and volume of dose.
Authors should describe about then in more detail.
Results
How did the changes of body weight?
Authors should add the data of changes of body weight in all groups.
Figure 4: Why didn't authors detect the target protein and the housekeeping protein on the same membrane? There were only contains the results for separate membranes the target protein and the housekeeping protein in WB bands file. Furthermore, all bands of Figure 4 were not existed in original images. Authors should use from original images.
Figure 6: (D) There was no yellow arrow in the photo nevertheless authors described about it in Figure legends. Authors should add a yellow arrow.
Authors described A-CAR and/or L-CoQ10 hepatoprotection might be a reduction of oxidative stress, inflammation, and apoptosis.
However, authors only measured caspase-3 activity. Did authors observe apoptotic hepatocytes in liver specimen of PRA-intoxicated group? Authors should detect apoptosis using TUNEL method using liver specimens.
Authors should clarify the mechanism of hepatoprotection by A-CAR and/or L-CoQ10 in more detail.
Author Response
Reviewer 1:
Comments and Suggestions for Authors
This study demonstrated that “Acetyl-L-carnitine and liposomal co-enzyme Q10 attenuate hepatic inflammation, apoptosis, and fibrosis induced by propionic acid”. This manuscript is very interesting, however, there are several concerns relating to that should be carefully addressed by the authors.
Point by point reply to reviewer 1:
Thank you for your valuable comments, much appreciated.
Materials and Methods
- How did authors decide the concentrations of PRA (250 mg/kg/day), A-CAR (100 mg/kg/day) and L-CoQ10(10 mg/kg/day) for dose to rats?
Reply: All the references were written in the experimental design.
The drug doses have been selected based on previous studies as follow:
Propionic acid dose (Paudel, et al., 2020), reference number 48.
Acetyl-L-carnitine dose (Bielefeld, et al., 2008), reference number 49.
Co-enzyme Q10 (El-Sheikh, et al., 2012), reference number 50.
- Authors described as “All treatments were given along with PRA for five days.”
All samples were orally administered; however, authors did not describe timing of administration and volume of dose. Authors should describe them in more detail.
Reply: All details are added in the experimental design, lines (280-285) as follow:
Rats were divided into five groups, six rats each. Group I (control): rats were received 1% carboxymethylcellulose; Group II: rats were orally intoxicated with PRA at a dose of 250 mg/kg/day [48]; Group III: rats were treated with A-CAR 100 mg/kg/day [49] one-hour post PRA administration; Group IV: rats were orally administered 10 mg/kg/day of L-CoQ10 [50] one hour after PRA intoxication; Group V: PRA‐intoxicated rats were orally administered a mixture of A-CAR and L-CoQ10 at same previous doses one hour post PRA administration. All treatments were given at volume of 0.5 ml along with PRA for five days.
Results
- How did the changes of body weight? Authors should add data of changes of body weight in all groups.
Reply: We didn’t notice any changes in body weight, as the treatment was only 5 days.
Also, there was no expected change in body weight between groups during this period.
- Figure 4: Why didn't authors detect the target protein and the housekeeping protein on the same membrane? There were only contains the results for separate membranes the target protein and the housekeeping protein in WB bands file. Furthermore, all bands of Figure 4 were not existed in original images. Authors should use original images.
Reply: Done, the bands for all target proteins and housekeeping proteins were amended and replaced from the original images.
Regard to the WB membrane: Doing beta actin with our target protein at the same time might show cross reactivity and produce unwanted signals, and also our detection system only capture an image before signal saturation thus we prefer to detect each target separately.
- Figure 6: (D) There was no yellow arrow in the photo nevertheless authors described it in Figure legends. Authors should add a yellow arrow.
Reply: Yes, the sentence has been deleted from the figure legend.
- Authors described A-CAR and/or L-CoQ10 hepatoprotection as a reduction of oxidative stress, inflammation, and apoptosis. However, authors only measured caspase-3 activity. Did authors observe apoptotic hepatocytes in liver specimen of PRA-intoxicated group? Authors should detect apoptosis using TUNEL method using liver specimens.
Reply: In our study, we measured the activity level of the cleaved caspase-3 using ELISA, analyzed the expression level of cytokeratin 18, and finally we detected necrotic areas using H &E stains.
As caspase-3 is responsible for most of the proteolysis during apoptosis, so the detecting of cleaved caspase-3 is considered a reliable marker for cell death. Also, cytokeratin18 represents approximately 5% of the total protein in the liver and plays an important function in apoptosis, thus can serve as a marker of cell death. During apoptotic processes, CK-18 is released from the hepatic cell, and it is cleaved by effector caspases that facilitate the formation of apoptotic bodies and augment the apoptotic signal.
- Authors should clarify the mechanism of hepatoprotection by A-CAR and/or L-CoQ10 in more detail.
Reply: The details of the mechanism of hepatoprotection by A-CAR and/or L-CoQ10 have been added in the discussion lines (176-181; 192-196; 200-209; 236-243).

Reviewer 2 Report
Dear the Editor
Alhusaini AM et al reported the protective role of acetyl-L-carnitine and CoQ10 on propionic acid-induced liver injury in rats. These authors demonstrated an elevation of liver enzymes (Fig. 1), which was inversely associated with biomarkers for oxidative stress (Fig. 2). Under this experimental setting, propionic acid exibited a negative effect on inflammation (Fig. 3). Similar effect was observed in the expression of TGFb and SMAD (Fig. 4). This finding was further supported by changes in other biomarkers such as HMGB1 and CK18 expression and histological observation (Figs 5&6). This is a well-organized paper emphasizing the protective role of these antioxidants in propionic acid-induced liver injury.
Major concerns:
1) Did these authos examine other enzyme activity such as catalase and glutathione peroxidase?
2) Given that the activity of caspase-3 was elevated, which cells were susceptible to apoptosis in this animal model?
3) Please discuss possible changes in ascorbic acid (vitamin C) in this animal model.
4) In discussion, these authors discussed the role of SMAD3 specifically. In Fig. 4, please specify which SMAD was examined. Is this also SMAD3 as indicated in Fig legend?
Minor concerns:
1) Please correct typos in text. For example:
a) L39, chemicals detoxification seemed to be chemical detoxification.
b) L156, liver Injury seemed to be liver injury.
Author Response
Reviewer 2:
Comments and Suggestions for Authors
Dear the Editor
Alhusaini AM et al reported the protective role of acetyl-L-carnitine and CoQ10 on propionic acid-induced liver injury in rats. These authors demonstrated an elevation of liver enzymes (Fig. 1), which was inversely associated with biomarkers for oxidative stress (Fig. 2). Under this experimental setting, propionic acid exhibited a negative effect on inflammation (Fig. 3). Similar effects were observed in the expression of TGFb and SMAD (Fig. 4). This finding was further supported by changes in other biomarkers such as HMGB1 and CK18 expression and histological observation (Figs 5&6). This is a well-organized paper emphasizing the protective role of these antioxidants in propionic acid-induced liver injury.
Point by point reply to reviewer 2:
Thank you for your valuable comments, much appreciated.
Major concerns:
- Did these authors examine other enzyme activity such as catalase and glutathione peroxidase?
Reply: In fact, we didn’t examine catalase and glutathione peroxidase, we only examined the level of the end-product of lipid peroxidation (MDA), the activity of SOD, and the level of glutathione (as indicators for oxidative stress).
- Given that the activity of caspase-3 was elevated, which cells were susceptible to apoptosis in this animal model?
Reply: Apoptosis is mediated by caspases enzymes, which trigger cell death by cleaving specific proteins in the cytoplasm and nucleus. Liver cells, especially hepatocytes and cholangiocytes, are particularly susceptible to death receptor-mediated apoptosis.
- Please discuss possible changes in ascorbic acid (vitamin C) in this animal model.
Reply: Ascorbic acid was not used in our study.
4) In discussion, these authors discussed the role of SMAD3 specifically. In Fig. 4, please specify which SMAD was examined. Is this also SMAD3 as indicated in Fig legend?
Reply: Done, SMAD has been changed to SMAD3 in Fig.4
5- Minor concerns:
1) Please correct typos in text. For example:
- a) L39, chemicals detoxification seemed to be chemical detoxification.
- b) L156, liver Injury seemed to be liver injury.
Reply: Done.

Round 2
Reviewer 1 Report
Authors almost responded for my questions and comments.
Reviewer 2 Report
Revised manuscript addressed all concerns raised by Reviewer.